# Impact of Corneal Crosslinking on Endothelial and Biomechanical Parameters in Keratoconus

**DOI:** 10.3390/jcm14134489

**Published:** 2025-06-25

**Authors:** Maria-Silvia Dina, Maria-Cristina Marinescu, Cătălina-Gabriela Corbu, Mihaela-Monica Constantin, Cătălina-Ioana Tătaru, Călin-Petru Tătaru

**Affiliations:** 1Doctoral School, Carol Davila University of Medicine and Pharmacy, 020021 Bucharest, Romania; maria.marinescu@drd.umfcd.ro; 2Oftaclinic Ophthalmology Clinic, 040254 Bucharest, Romania; carol_corbu@yahoo.com (C.-G.C.); mihaelamonicaconstantin@yahoo.com (M.-M.C.); 3Ophthalmology Department, “Carol Davila” University of Medicine and Pharmacy, 020021 Bucharest, Romania; catalina-ioana.tataru@umfcd.ro (C.-I.T.); calin.tataru@umfcd.ro (C.-P.T.)

**Keywords:** keratoconus, corneal crosslinking, corneal endothelium, corneal biomechanics

## Abstract

**Background/Objectives**: Keratoconus (KC) is a corneal ectatic disease, characterized by the progressive thinning of the cornea, myopia, and astigmatism, which lead to a decrease in visual acuity. Corneal collagen crosslinking (CXL) is an efficient method of stopping the progression of the disease. The objective of this study is to investigate the endothelial and biomechanical properties of the cornea in keratoconus patients, before and after undergoing corneal collagen crosslinking. **Methods**: A total of 66 eyes were diagnosed with progressive keratoconus and were recommended epi-off corneal crosslinking. Before the procedure, they were investigated with corneal topography (for minimum, maximum, average keratometry, and corneal astigmatism), specular microscopy (for the following endothelial cell parameters: number, density, surface, variability, and hexagonality), and an ocular response analyzer (for the following biomechanical parameters: corneal hysteresis and resistance factor). All measurements were repeated 1 month and 6 months after the intervention. **Results**: Several parameters differ according to the Amsler–Krumeich stage of keratoconus: in more advanced stages, patients present higher endothelial cell variability, a lower number of endothelial cells in the paracentral region of the cornea, lower CCT and CRF, and higher keratometry and astigmatism. Endothelial cell variability and number correlate with average keratometry, and there are also strong correlations between topography and CH and CRF. After CXL, the paracentral number of endothelial cells decreased; cell variability and average cell surface increased. **Conclusions**: More advanced keratoconus cases present with altered corneal biomechanics and topographical parameters, the endothelial layer also being affected proportional to the stage of the disease and also slightly affected after corneal collagen crosslinking.

## 1. Introduction

Keratoconus (KC) is a corneal ectatic disease, characterized by a progressive thinning of the cornea, myopia, and astigmatism, which lead to a decrease in visual acuity [1]. This disease is often found in younger people, appearing during puberty, progressing throughout the second and third decades, and stabilizing around the age of 40 years old [2]. Incidence is estimated to be in the range of 1.5–25 cases per 100,000 persons/year, with a higher rate in Middle Eastern and Asian patients [3].

One significant step in the treatment of keratoconus was represented by corneal collagen crosslinking (CXL), introduced in clinical practice in 2003, which halts ectasia progression [4]. CXL aims to stabilize corneal ectasia, being a surgical procedure that is used to increase the biomechanical strength of the cornea by forming chemical bonds between collagen fibrils, which leads to the inhibition of keratoconus progression, therefore maintaining visual acuity over time and preventing complications [5,6,7].

The epithelium-off CXL technique was first applied to human keratoconus corneas in 1998 and consists of epithelial removal, after which the corneal stroma is saturated with the photosensitizer riboflavin (vitamin B2) for 30 min. Then, ultraviolet A radiation (UVA, 370 nm) is applied, which produces reactive oxygen species, which in turn induce interfibrillar collagen crosslinks [8]. The reason for the removal of the epithelium has been described as allowing the adequate penetration of riboflavin into the stromal tissue, where it absorbs UVA light and produces actual crosslinking between collagen fibrils in the corneal stroma. The disadvantage of epithelial removal is that it causes significant pain and discomfort in the first few days postoperatively, in addition to a 3–8% chance of epithelial healing problems [9,10].

Specular microscopy is a noninvasive technique that allows us to visualize and analyze the corneal endothelium. Corneal endotheliopathy is a broad term used to classify several diseases and clinical conditions that affect the structure and function of the corneal endothelium. One of the physiological functions of the endothelium is to secrete a collagen matrix that forms the Descemet membrane, but the primary physiological function of the corneal endothelium is to maintain the health and transparency of the corneal stroma [11].

Corneal biomechanical properties present more and more significance in ocular pathology, such as glaucoma [12] and refractive errors (myopia [13] and hyperopia [14]). An Ocular Response Analyzer (ORA) is a device that measures two essential biomechanical parameters: corneal hysteresis (CH) and the corneal resistance factor (CRF). In keratoconus patients, studies have shown that CH and CRF values are statistically significantly lower compared to healthy individuals, and the values are even lower the more advanced the keratoconus is. Also, the CRF value is lower compared to the CH value in keratoconus patients, so the difference between CH and CRF is positive and increases as the condition progresses [15].

The objective of this study is to investigate the endothelial and biomechanical properties of the cornea in keratoconus patients, before and after undergoing corneal collagen crosslinking.

## 2. Materials and Methods

This study is designed as a retrospective, non-randomized, interventional study. All patients who presented at the Oftaclinic Clinic in Bucharest between 2019 and 2022 were screened for inclusion in this study. This study’s cohort was formed by applying the inclusion and exclusion criteria. All patients, and, in the case of minors, their legal representatives, offered informed consent. This study was approved by the Oftaclinic Ethics Committee (Approval 2/4 February 2019).

The inclusion criterion was the diagnosis of progressive keratoconus. Keratoconus was diagnosed and staged according to Amsler–Krumeich staging, and progression was defined as an increase of at least 1 diopter (D) within a year of follow-up of the steepest corneal meridian (Kmax), decreased visual acuity, or thinning of the cornea by more than 5% of the corneal thickness [16].

Patients were excluded from this study if they presented other ocular pathologies (vitreoretinal pathology, glaucoma, ocular hypertension, cataract, and history of corneal surgery or corneal disease—particularly Fuch’s endothelial dystrophy), if they were pregnant or breastfeeding, or if they were not compliant with the investigations needed (for example, low waveform score in Ocular Response Analyzer testing). All patients had normal intraocular pressure (under 21 mmHg) both at the first consultation and at the consultations following the intervention.

Furthermore, patients were excluded if they presented any of the following known CXL contraindications: known sensitivity against ingredients used during the procedure, corneal thickness under 400 microns, and history of corneal pathology, such as herpes simplex keratitis [17].

All patients underwent a complete ocular consultation, after being instructed to stop wearing soft contact lenses for at least 2 weeks or rigid contact lenses for at least 4 weeks before the consultation [18]. During the consultation, the following assessments were also included:-Aladdin corneal topography (Topcon, Tokyo, Japan)—to determine maximum, minimum, average keratometry, and corneal astigmatism.-Specular microscopy (Nidek, Gamagori, Japan) for the determination of the following corneal endothelial parameters: number of cells, cell density, average cell surface, cell variability, and hexagonality—these parameters being calculated both centrally and paracentrally, and central corneal thickness.-Ocular Response Analyzer (ORA) (Reichert Ophthalmic Instruments Inc., Depew, NY, USA)—in order to determine the following corneal biomechanical properties: corneal hysteresis (CH) and corneal resistance factor (CRF). The Goldmann-correlated intraocular pressure calculated with the ORA over 21 represented an exclusion criterion.

Topographic evaluation and specular microscopy were performed after hydrating the ocular surface, considering that patients with keratoconus have a shorter tear film break-up time compared to the general population [19].

Patients were classified according to the Amsler–Krumeich classification of keratoconus (one criterion being needed to assign a certain stage to the eye), which accounted for myopia and astigmatism degree, mean keratometry, and minimal central corneal thickness [20].

The crosslinking procedure was performed in cases of progressive keratoconus, following the inclusion and exclusion criteria [21].

The crosslinking intervention followed the standard protocol. Stopping the progression of keratoconus was achieved by photo-oxidative crosslinking “epi-off” technique, a method that involves epithelial removal. In order to perform the procedure, local topical anesthesia is required, which does not influence the surgical procedure. After administering anesthesia and cleaning the regional tissue with Betadine (Povidone Iodate), a sterile field was applied. A speculum was applied to keep the eyelids open. After epithelial removal mechanically, the corneal stroma was saturated with riboflavin (vitamin B2) 0.1% in 20% dextran by photosensitization for a certain period of time (1 drop every 2 min for 30 min). Subsequently, ultraviolet A radiation (UVA, 370 nm) was applied for 30 min, continuing the infusion of riboflavin at 2 min (the “Dresden” technique) [21,22].

After the procedure, the patient was fitted with a therapeutic contact lens for 7 days, which plays a role in pain relief and the mechanical protection of the cornea [23].

Postoperative treatment consisted of the application of anti-inflammatory drops, topical antibiotics, and artificial tears for 14 days, which play a role in pain relief and the prevention of inflammation [23,24].

The data analyzed within this study originate from the pre-intervention consultation and from the consultations performed 1 month and 6 months after the CXL intervention.

The data analyzed in this article were tested with Levene’s Test, followed by the dependent *t* test, in order to identify significant differences between examinations (same patient—before the intervention, and 1 month and 6 months after the intervention). The 4 keratoconus stages were compared using the One-way ANOVA test, followed by the post hoc analysis Tukey test.

Pearson’s correlation coefficient (“Pearson’s r”) was calculated to determine the degree of correlation between variables (weak correlation: Pearson’s r between 0.3 and −0.3, moderate correlation: 0.3–0.5 or −0.3–−0.5, and strong correlation: over 0.5, under −0.5). The *p* value of 0.05 was considered the threshold for statistical significance. Statistical analysis was performed using the Statistical Package IBM SPSS Statistics for Windows, version 26 (IBM Corp., Armonk, NY, USA).

## 3. Results

### 3.1. Descriptive Results of the Entire Cohort

This study includes 47 patients, which provided 66 eyes for analysis (diagnosed with progressive keratoconus and recommended corneal crosslinking). A total of 17.02% of patients were female and 82.98% were male. The average age was 25.40 years old (SD 6.68). According to the Amsler–Krumeich classification, 48.48% of eyes were stage I keratoconus (32 eyes), 33.33% stage II (22 eyes), 12.12% stage III (8 eyes), and 6.06% stage IV (4 eyes). The average values of the parameters followed are in Table 1, before the procedure and after 1 and 6 months.

When analyzing the entire cohort, the endothelial layer was only minimally modified after CXL: cell variability increased significantly at 1 month after CXL (*p* 0.003), and more changes emerged at 6 months: cell variability was still significantly higher than before the procedure (*p* 0.001), average cell size was significantly higher (*p* 0.014), and the paracentral number of cells was significantly lower (*p* 0.038) (Figure 1) (Table 1).

The endothelial parameters did not differ significantly between the different KC stages: as the ectasia was more advanced, the cell variability increased (*p* value 0.033) (mean value of 24.81% in stage I, 24.77% in stage II, 26.00% in stage III, and 28.75% in stage IV); however, hexagonality was not significantly modified (67.19% in stage I, 69.96% in stage II, 65.50% in stage III, and 71.25% in stage IV). However, other parameters differed significantly: CRF was smaller (*p* value 0.006) and the central cornea was thinner (*p* < 0.001). Also, topographical parameters were more advanced, with higher minimum, maximum, and average keratometry and astigmatism (<0.001) in more advanced keratoconus cases (see Table 2).

### 3.2. Stage I Keratoconus Group—Evolution After Corneal Crosslinking

The stage I keratoconus group was composed of 32 eyes—all mean values are presented in Table 3. The paired T-test, comparing pre-CXL variables with post-CXL values at one and six months, reveals that very few parameters were modified after the procedure: the number of cells (central and paracentral), the hexagonality and paracentral cell density are significantly decreased at 6 months compared to the baseline, and cell variability is significantly increased both at 1 and at 6 months compared to the baseline—see Table 4.

### 3.3. Stage II Keratoconus Group—Evolution After Corneal Crosslinking

The stage II keratoconus group was composed of 22 eyes—all mean values are presented in Table 5. The paired T-test, comparing pre-CXL variables with post-CXL values at one and six months, reveals that few parameters were modified after the procedure, and the differences become significant only after 6 months (See Table 6). Cell density is significantly decreased at 6 months compared to the baseline; cell average surface area and variability are significantly increased at 6 months compared to the baseline. Furthermore, both Kmax and Kaverage are significantly lower at 6 months post-procedure.

### 3.4. Stage III Keratoconus—Evolution After Corneal Crosslinking

The stage III keratoconus group was composed of 8 eyes—all mean values are presented in Table 7. The paired T-test, comparing pre-CXL variables with post-CXL values at one and six months, reveals that CRF significantly improves after 1 month, and paracentral hexagonality is affected only after 6 months (See Table 8).

### 3.5. Stage IV Keratoconus—Evolution After Corneal Crosslinking

The stage IV keratoconus group was composed of 4 eyes—all mean values are presented in Table 9. The paired T-test, comparing pre-CXL variables with post-CXL values at one and six months, reveals only a modified CRF average at one month after the procedure (see Table 10).

### 3.6. Correlations in the Entire Cohort

Correlations were calculated between endothelial, biomechanical, and topographical parameters, in the initial pre-crosslinking cohort. However, no significant correlations were found between either of the endothelial parameters or the biomechanical parameters. On the other hand, there were several statistically significant correlations between paracentral endothelial parameters and topographic variables in the initial cohort. There are significant, weak, positive correlations between paracentral cell variability and either average keratometry and minimum keratometry. There are also significant, moderate, negative correlations between the paracentral number of endothelial cells and either maximum, minimum, or average keratometry—see Table 11.

Also, as expected, there are significant correlations between the biomechanical parameters and topographic parameters—CH has negative, moderate correlations with minimum, maximum, and average keratometry; CRF has negative, moderate correlations with minimum, maximum, and average keratometry and astigmatism; and CCT has negative, moderate–strong correlations with minimum, maximum, and average keratometry—see Table 12, Figure 2.

### 3.7. Correlations Between Baseline Values and Post-CXL Changes

By correlating the degree of change in several variables to the baseline values, we can understand how crosslinking influences both biomechanical and endothelial data. The degree of change in several endothelial parameters depended on the baseline biomechanical and topographical characteristics of the cornea—see Table 13. Firstly, there was a weak–moderate positive correlation between age and cell variability increase after CXL—the older the patient, the greater the variability increase in the endothelial layer—see Figure 3.

There was a weak–moderate negative correlation between CH and Avg cell surface increase at 6 months, and a negative moderate correlation between CRF and CV increase at 1 month, revealing that the higher the biomechanical parameters, the lower the damage to the endothelial layer in terms of cell surface and variability. There were also weak–moderate positive correlations between Kavg and the increase in Avg cell surface and variability at 6 months—the steeper the cornea (e.g., the more advanced the corneal ectasia), the greater the endothelial damage in terms of cell surface and variability.

There were also significant correlations between the baseline endothelial layer characteristics and the degree of change in the endothelial layer—see Table 14. There was a weak negative correlation between CV and the number of paracentral cells lost at 6 months, and a weak positive correlation between CV and the increase in cell surface at 6 months—the lower the variability in the endothelial layer, the lower the endothelial damage in terms of cells lost and surface increase—see Figure 4.

## 4. Discussion

In this study, we identified several differences between keratoconus eyes, depending on how advanced the corneal ectasia was, although this study was conducted on a small number of patients. According to the Amsler–Krumeich classification, the higher the stage, the greater the corneal astigmatism and average keratometry, and the thinner the cornea, these significant differences being found in our cohort as well.

Sturbaum and Peiffer (1993) found that in the early stages of keratoconus, the endothelium appears normal [25]. As the condition progresses and the corneal stroma thins, the endothelial cells flatten. This likely reflects the stretching of the endothelial cells as they attempt to maintain continuity across the progressively ectatic posterior surface [26]. Furthermore, the endothelium may be damaged in keratoconus through Descemet membrane breaks, contact lens wear and excessive eye rubbing, and thin corneas, which allow a greater deal of UV damage [27]. Later in the course of the disease, endothelial changes include pleomorphism, polymegathism, endothelial cell degeneration, and exposure of fibrin, inflammatory cells, or both on the endothelial surface. Endothelial and Descemet membrane ruptures occur in 11% to 35% of keratoconic corneas, and the healing of these defects can also lead to pleomorphism and polymegathism [26]. While advancing endothelial cell loss may decrease visual acuity through the loss of stromal transparency, unfortunately there is no definitive treatment to halt the loss of cells, except corneal transplantation in advanced cases [28].

There is research in the literature hinting at an altered endothelial cell layer in keratoconus. A large study of over 700 eyes revealed that more advanced cases present lower cell density and a higher variability [29]. This is probably due to the small number of patients with advanced stages included in our study. These results are similar to ours, however only the variability was significant in our cohort. On the other hand, there are studies in the literature that found no differences in density, variability, or hexagonality between Amsler–Krumeich KC stages [30].

A study applying deep learning technologies has found that an average number of cells in keratoconus is 175.00, comparable but slightly higher than our value of approximately 137 found in the present study [31]. More importantly, this average differed significantly to that of myopic eyes, serving as a control group [31]. Another recent study included Egyptian keratoconus patients, with Amsler–Krumeich stages 1–3, and found a significant decrease in cell density and an increase in variability [27]. Compared with our cohort, these parameters revealed a more significant alteration in the endothelial layer, suggesting that more research is needed to understand the endothelial behavior of keratoconus in different populations and which factors may influence endothelial health. Importantly, in our cohort, the average central endothelial cell density was sufficient to maintain corneal transparency (over 2900 cells/mm^2^), and the decrease in cell density was non-significant following CXL.

Keratoconus is a disease that involves the degradation of collagen fibers. Histologically, KC is characterized by a lower number of collagen fibers, in conjunction with a thinner stroma, abnormal keratocytes, affected Bowman’s membrane, and basement membrane with iron deposits [32]. Increased expression of the matrix metalloproteinase (MMP-1) leads to extracellular matrix degradation, affecting collagen, and a decreased expression of an enzyme responsible for natural collagen crosslinking contributes to a weaker corneal structure in keratoconus [33], with the addition of a higher lacrimal concentration of inflammatory markers in keratoconus patients [34]. As collagen is responsible for the elasticity and resistance of the cornea, it is known that CH and CRF are lowered in keratoconus. A large comparative study found an average of 7.5 mmHg and 6.2 mmHg for CH and CRF, respectively, significantly lower than the 10.8 and 11.0 averages in normal eyes [35]. These values are comparable with the averages obtained in our study. Furthermore, as expected, keratoconus variables (average keratometry and corneal thickness) correlate well with CH and CRF, similar to other results in the literature [36].

A biomechanical evaluation of the cornea holds great value as focal alterations may appear before the topographical alterations become evident, therefore holding great potential for early ectasia diagnosis [37]. CH, and even moreso, CRF, present good accuracy in differentiating normal and ectatic corneas [37]. Similarly, in our study, only CRF, not CH, differed significantly between stages of keratoconus, supporting the role of diagnostic and staging in the variations in CRF.

Besides the variable relationship with the disease stage, in our cohort, endothelial properties correlated with other parameters specific to KC—namely corneal astigmatism and keratometry readings (minimum, maximum, and average). These findings have been previously suggested in the literature, with endothelial cell density being negatively correlated with the steepest and flattest corneal meridian and positively correlated with minimum corneal thickness [38].

Corneal crosslinking is a well-established treatment in keratoconus that is used to stop KC evolution by increasing the biomechanical strength of the cornea. Using confocal microscopy, the corneal changes that occur after CXL could be examined. This procedure leads to microscopic changes, such as the appearance of a demarcation line, the apoptosis of keratocytes, and stroma edema—however, these changes generally resolve within six months and are accompanied by visual acuity increase [39].

In our study, there were no statistically significant changes in terms of corneal topography or biomechanics, as the follow-up reported here was only up to 6 months. Studies reporting from 1 year up to 7 years of follow-up describe significant improvements in keratometry and refractive errors, however less so in biomechanical parameters [40,41]. On the other hand, certain endothelial alterations have been observed in our cohort, such as increased cell variability and average cell area, and decreased paracentral number of cells.

The literature states that increased cell variability is the first sign of cell damage, and the cell surface area increases with increasing CV [42]. Although in our cohort no patient had a coefficient higher than the normal limit, our study showed that a higher value of cellular variability before CXL was correlated with a decrease in the number of endothelial cells, but most importantly with an increase in cell surface after the procedure. A recent study that measured the effect of CXL on corneal endothelial cell morphology in patients with KC observed that the coefficient of variation was also statistically significantly higher at 6 months postoperatively than the initial value and there was a slight change in the percentage of hexagonal cells [43]. These observations can also be found in our cohort in stages I-III. Thus, although the corneal endothelium is not affected per se, the possibility of minor damage before CXL, suggested by the higher value of the coefficient of variation, which probably also increases after the CXL procedure, leads to an increase in the surface area of endothelial cells.

A study of 400 eyes of patients with keratoconus who underwent CXL and were followed for 4 years showed that the procedure stopped the progression of KC in all age groups, but the functional and morphological results were better in patients aged 18 to 39 years [44]. Similarly, our study showed a halt in the evolution of keratoconus after the crosslinking procedure in all patients, but the older the age, the greater the CV difference. On the other hand, it is known that in the healthy population, the corneal endothelial layer has a very high cell density at birth, which decreases with age at a rate of approximately 0.3–0.6% per year [45]. A study conducted on healthy volunteers aged 11 to 88 years in Thailand, which aimed to determine the influence of age on corneal endothelial thickness and morphology, showed a significant decrease in corneal thickness, cell density, and hexagonality, while endothelial cell surface area and cellular variation increased with age [46]. Although the age of the patients in our cohort does not exceed 50 years, the increase in the coefficient of endothelial variation after the crosslinking procedure at older ages could be explained by the overlap of some physiological processes.

Given the limitations of our study regarding the proportion of patients in stages III and IV compared to stages I and II, the correlation of corneal endothelial changes post-CXL at each stage is also limited. It is important, however, that the stages that included the largest number of patients, respectively I and II, showed morphological changes in endothelial cells that included a significant increase in the coefficient of variation and changes in the surface and hexagonal shape of endothelial cells at 6 months post-intervention. This is also found in a study comparing the effects of corneal crosslinking on the corneal endothelium in patients with KC compared to those with post-Lasik ectasia, which found that CV was statistically significantly higher 3 months after surgery compared to preoperative data in KC, and the percentage of hexagonal cells showed the smallest changes. In addition, similar to our study, a decrease in cell density occurred 6 months after CXL, which in their study recovered after one year [47].

A study similar to ours, with a follow-up of one year, has not found any differences in terms of cell density; however, it has not investigated other endothelial parameters that we found to be different before and after CXL [48]. Another recent study revealed a statistically significant decrease in cell density after 6 months; however, the final density did not impede endothelial function and corneal clarity [49]. The significant variation in endothelial parameters in the paracentral area after crosslinking obtained in our study could also be explained by the fact that the paracentral area in the patients in this study is the most deformed area and has the lowest corneal resistance (it coincides with the area of corneal ectasia). While these modifications are statistically significant, it is unlikely that they are clinically significant. The usage of riboflavin prevents UVA damage to endothelial cells, and more studies have reported no density decrease and far fewer have reported a decrease in endothelial cell density after CXL [50]. Most likely, significant changes occur in very thin keratoconic corneas, where the endothelium experiences a higher dose of UVA [51]. The UVA reaction from the crosslinking process damages subbasal nerves [47,52], which are essential in the endothelial pump. In this way, changes occur in the endothelial cells. Over time, endothelial cells that have not been affected tend to replace damaged cells [50].

In our study, a significant increase in CRF was also found in patients with stage III AK at 1 month post-CXL. However, this is inconsistent with stage IV. In any case, the small number of subjects in stages III and IV AK is a limitation of our study; therefore, more research is needed to determine if the biomechanical changes occurring post-CXL are more evident in more advanced ectatic corneas. Current studies on this topic are still under debate, with some studies finding no changes in either CH or CRF [53], and others showing increases in these parameters [54].

The methodology chosen for this study was a non-randomized one, because the patients followed treatment according to the clinical indication and could not be randomized, the only method to stop the evolution of keratoconus nowadays being the corneal crosslinking method. Otherwise, without the application of this treatment, with the progression of the disease, severe damage to the structure of the cornea leads to the need for a corneal transplant through different techniques (Bowman layer transplantation, deep anterior lamellar keratoplasty, or penetrating keratoplasty), techniques that bring with them the risk of damage to the endothelial layer, unrelated to keratoconic damage as much as the procedure itself [55,56,57].

Another significant limitation of this study is the minimal clinical data we have on our patients: no further data are known regarding atopic background or behavior, such as eye rubbing, which are known KC risk factors [58]. Furthermore, no data regarding visual acuity or method of visual correction (glasses, or rigid or soft contact lenses) are analyzed in this study, while it is known that, more often, visual correction in early KC stages is performed using spectacles or soft contact lenses, and in more advanced cases rigid lenses are more successfully used [59]. More research is needed through expanding the cohort of advanced KC (stage IV) and analyzing the influence of more factors on the progression of the disease and the response to corneal crosslinking.

## 5. Conclusions

In more advanced keratoconus, there is a decrease in the number of paracentral endothelial cells and an increase in variability, and a correlation between the characteristics of the endothelial layer and KC specific parameters—minimum, maximum, and average keratometry. This cellular layer is also affected after corneal crosslinking, the differences not being, however, clinically impactful.

More research is needed to determine if different corneal crosslinking protocols have a different effect on the endothelial layer, and also if the long-term follow-up of these patients would reveal a clinical significance of these endothelial alterations.

## Figures and Tables

**Figure 1 jcm-14-04489-f001:**
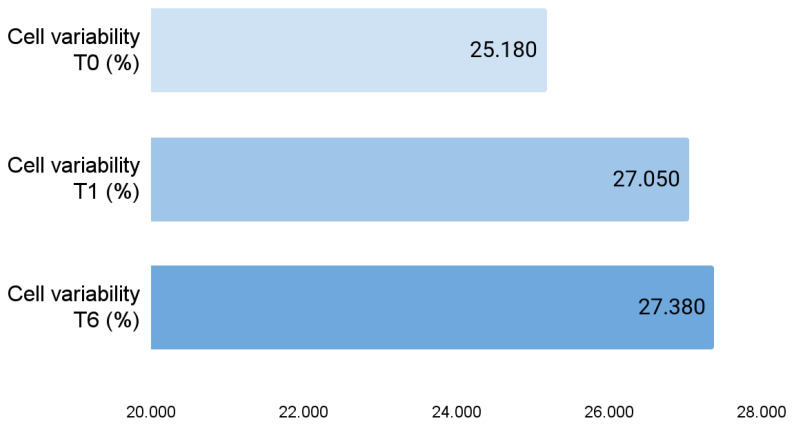
Bar charts representing average endothelial cell variability before CXL, after 1 month, and after 6 months.

**Figure 2 jcm-14-04489-f002:**
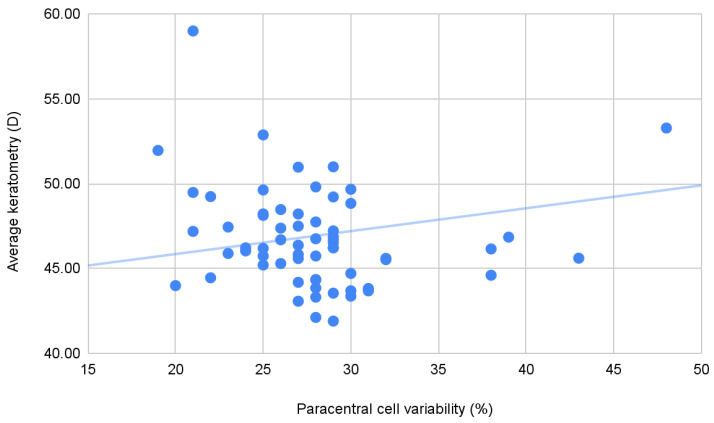
Correlations between average keratometry and paracentral cell variability in the entire study cohort before crosslinking.

**Figure 3 jcm-14-04489-f003:**
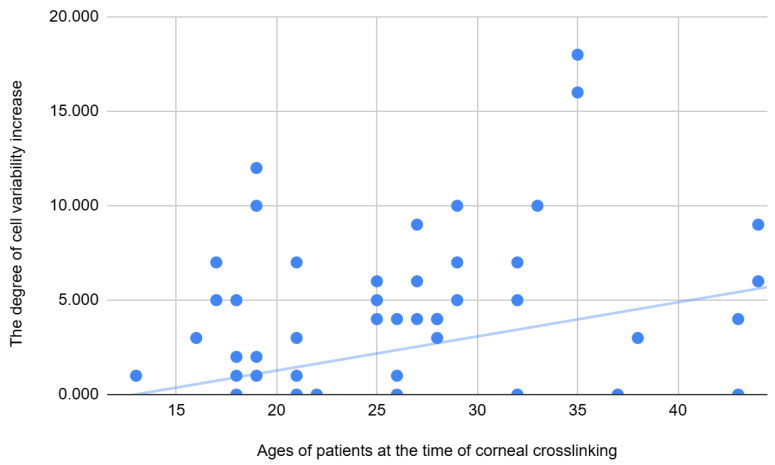
Correlations between the age of the patients and cell variability increase at 6 months after the procedure.

**Figure 4 jcm-14-04489-f004:**
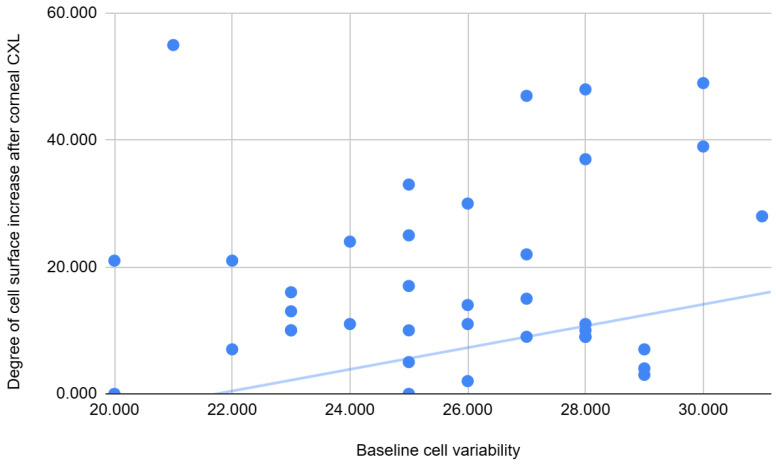
Correlations between the baseline cell variability and the degree of cell surface increase after CXL.

**Table 1 jcm-14-04489-t001:** Mean values and standard deviation (SD) in the entire cohort before (T0), 1 month after CXL (T1), and 6 months after CXL (T6). Bold items signify statistically significant differences compared to the baseline value.

		Mean ± SD			Mean ± SD
Number of cells	T0	136.86 ± 48.38	Number of cells (paracentral)	T0	1110.45 ± 406.01
	T1	140.97 ± 52.48		T1	1060.82 ± 487.53
	T6	127.30 ± 46.74		**T6**	**1003.98 ± 429.42**
Cell density (cell/mm^2^)	T0	2941.02 ± 281.53	Cell density (paracentral) (cell/mm^2^)	T0	2927.65 ± 253.68
	T1	2950.27 ± 264.00		T1	2932.82 ± 258.00
	T6	2914.53 ± 276.43		T6	2913.44 ± 247.38
Cell variability (%)	T0	25.18 ± 2.76	Cell variability (paracentral) (%)	T0	28.11 ± 5.36
	**T1**	**27.05 ± 4.54**		T1	29.70 ± 9.57
	**T6**	**27.38 ± 5.20**		T6	29.38 ± 7.41
Cell hexagonality (%)	T0	68.15 ± 6.18	Cell hexagonality (paracentral) (%)	T0	66.50 ± 4.31
	T1	67.23 ± 5.47		T1	66.94 ± 4.49
	T6	67.11 ± 6.19		T6	66.97 ± 4.33
Average cell surface (micrometers)	T0	342.73 ± 35.11	Central corneal thickness (μm)	T0	496.27 ± 36.06
	T1	341.82 ± 32.43		T1	495.26 ± 40.99
	**T6**	**348.61 ± 33.50**		T6	495.86 ± 40.29
Corneal hysteresis (mmHg)	T0	8.75 ± 1.37	Corneal resistance factor (mmHg)	T0	7.82 ± 1.74
	T1	8.75 ± 1.73		T1	7.83 ± 1.83
	T6	8.76 ± 1.95		T6	7.89 ± 2.34
Flattest meridian—Kmin (D)	T0	45.05 ± 3.39	Steepest meridian—Kmax (D)	T0	49.51 ± 4.88
	T1	45.10 ± 3.52		T1	49.56 ± 4.92
	T6	45.21 ± 3.48		T6	49.50 ± 4.82
Average keratometry—Kavg (D)	T0	47.50 ± 4.05	Corneal astigmatism (D)	T0	4.57 ± 2.99
	T1	47.32 ± 4.03		T1	4.50 ± 2.90
	T6	47.34 ± 3.92		T6	4.29 ± 3.04

**Table 2 jcm-14-04489-t002:** Mean values of significantly different parameters between keratoconus Amsler–Krumeich stages (before corneal collagen crosslinking).

KC Stages		CV	CRF	CCT	Kavg	Corneal Astigmatism
Stage I	Mean ± SD	24.81 ± 2.66	8.55 ± 1.62	507.31 ± 35.25	44.94 ± 1.45	2.52 ± 1.31
Stage II	Mean ± SD	24.77 ± 3.023	7.44 ± 1.65	494.18 ± 27.59	48.48 ± 2.26	4.97 ± 2.02
Stage III	Mean ± SD	26.00 ± 1.309	6.76 ± 1.57	494.37 ± 10.01	49.39 ± 3.25	8.98 ±0.96
Stage IV	Mean ± SD	28.75 ± 1.50	6.42 ± 1.13	423.25 ± 36.35	58.82 ± 3.91	9.57 ± 3.40

**Table 3 jcm-14-04489-t003:** Mean values and standard deviation (SD) in stage I KC, before CXL, and after 1 and 6 months. NrCells = number of cells, CD = cell density, Avg = average cell surface, CV = cell variability, Hex = cell hexagonality, CCT = central corneal thickness, NrCellsPC = number of paracentral cells, CDPC = paracentral cell density, CVPC = paracentral cell variability, HexPC = paracentral hexagonality, CH = corneal hysteresis, CRF = corneal resistance factor, Kmin = minimum keratometry, Kmax = maximum keratometry, Kavg = average keratometry, and Astigm = astigmatism. Bold items signify statistically significant differences compared to the baseline value.

	Mean ± SD		Mean ± SD
Endothelial variables (both central and paracentral)			
NrCells	137.00 ± 48.7	NrCellsPC	1130.62 ± 452.2
NrCells I	136.53 ± 48.42	NrCellsPC I	1053.50± 502.7
**NrCells6**	**120.06 ± 43.4**	**NrCellsPC6**	**943.00 ± 449.6**
CD	2909.87 ± 312.8	CDPC	2939.25 ± 267.8
CD I	2956.50 ± 289.4	CDPC I	2950.53 ± 290.8
CD6	2915.53 ± 313.6	**CDPC6**	**2904.81 ± 272.4**
CV	24.81 ± 2.7	CVPC	28.75 ± 4.4
**CV I**	**28.69 ± 5.1**	CVPC I	29.59 ± 6.2
**CV6**	**28.19 ± 5.4**	CVPC6	29.69 ± 4.6
Hex	67.19 ± 6.9	HexPC	65.28 ± 4.6
Hex I	65.56 ± 6.3	HexPC I	65.34 ± 5.1
Hex6	65.00 ± 6.8	HexPC6	65.63 ± 5
Avg	346.56 ± 39.9	CCT	507.31 ± 35.2
Avg I	341.69 ± 36	CCT I	509.03 ± 38
Avg6	347.62 ± 39.5	CCT6	507.75 ± 38.2
Biomechanical corneal properties			
CH	9.12 ± 1.3	CRF	8.54 ± 1.6
CH I	9.08 ± 1.4	CRF I	8.33 ± 1.6
CH6	9.03 ± 2.3	CRF6	8.51 ± 2.6
Topographical properties			
Kmin	43.68 ± 1.5	Kmax	46.2 ± 1.7
Kmin I	43.79 ± 1.7	Kmax I	46.45 ± 2
Kmin6	44.0 ± 2	Kmax6	46.46 ± 2.3
Kavg	44.94 ± 1.45	Astigm	2.52 ± 1.3
KavgI	45.12 ± 1.7	AstigmI	2.66 ± 1.5
Kavg6	45.23 ± 1.9	Astigm6	2.47 ± 2.06

**Table 4 jcm-14-04489-t004:** Significant differences in parameters before CXL, after CXL at 1 and 6 months, and *p* value of difference in stage I KC. NrCells = number of cells, CV = cell variability, Hex = cell hexagonality, NrCellsPC = number of paracentral cells, and CDPC = paracentral cell density.

	Mean Difference (After CXL—Baseline) (*p* Value)
NrCells at 6 months vs. Baseline NrCells	−16.94 (0.009)
CV at 1 month vs. Baseline CV	3.87 (<0.001)
CV at 6 months bs. Baseline CV	3.37 (0.002)
Hex at 6 months vs. Baseline Hex	−2.19 (0.017)
NrCellsPC at 6 months vs. Baseline NrCellsPC	−187.62 (0.013)
CDPC at 6 months vs. Baseline CDPC	−34.437 (0.011)

**Table 5 jcm-14-04489-t005:** Mean values and standard deviation (SD) in stage II KC before CXL, and after 1 and 6 months. NrCells = number of cells, CD = cell density, Avg = average cell surface, CV = cell variability, Hex = cell hexagonality, CCT = central corneal thickness, NrCellsPC = number of paracentral cells, CDPC = paracentral cell density, CVPC = paracentral cell variability, HexPC = paracentral hexagonality, CH = corneal hysteresis, CRF = corneal resistance factor, Kmin = minimum keratometry, Kmax = maximum keratometry, Kavg = average keratometry, and Astigm = astigmatism. Bold items signify statistically significant differences compared to the baseline value.

	Mean ± SD		Mean ± SD
Endothelial variables (both central and paracentral)			
NrCells	135.91 ± 54.6	NrCellsPC	1125.36 ± 422.1
NrCells I	148.27 ± 62.3	NrCellsPC I	1035.09 ± 532.1
NrCells6	143.0 ± 50.5	NrCellsPC6	1086.18 ± 484.5
CD	2941.9 ± 296.2	CDPC	2908.32 ± 266.7
CD I	2894.86 ± 272	CDPC I	2866.45 ± 255.6
**CD6**	**2873.54 ± 267.1**	CDPC6	2876.23 ± 246.6
CV	24.77 ± 3.0	CVPC	26.77 ± 2.8
CV I	25.73 ± 3.1	CVPC I	29 ± 8.5
**CV6**	**26.5 ± 4.5**	CVPC6	28.09 ± 4.8
Hex	69.96 ± 5.2	HexPC	68 ± 3.9
Hex I	69.04 ± 3.9	HexPC I	68.5 ± 3.4
Hex6	69.86 ± 4.2	HexPC6	68.09 ± 3.3
CCT	494.18 ± 27.6	Avg	343.36 ± 35.6
CCT I	484.91 ± 39	Avg I	348.4 ± 32.9
CCT6	485.05 ± 38	**Avg6**	**352.73 ± 31.1**
Biomechanical corneal properties			
CH	8.432 ± 1.446	CRF	7.441 ± 1.646
CH I	8.718 ± 2.059	CRF I	7.586 ± 2.151
CH6	8.755 ± 1.122	CRF6	7.696 ± 1.532
Topographical properties			
Kmin	45.995 ± 2.739	Kmax	50.962 ± 2.183
Kmin I	45.812 ± 2.575	Kmax I	50.860 ± 2.415
Kmin6	45.725 ± 2.452	**Kmax6**	**50.614 ± 2.086**
Kavg	48.481 ± 2.262	Astigm	4.967 ± 2.017
KavgI	48.301 ± 2.318	AstigmI	5.116 ± 1.970
**Kavg6**	**48.169 ± 2.110**	Astigm6	4.890 ± 1.708

**Table 6 jcm-14-04489-t006:** Significant differences in parameters before CXL, after CXL at 1 and 6 months, and *p* value of difference in stage II KC. CD = cell density, Avg = average cell surface, CV = cell variability, Kmax = maximum keratometry, and Kavg = average keratometry.

	Mean Difference (After CXL—Baseline) (*p* Value)
CD at 6 months vs. Baseline CD	−68.36 (0.026)
Avg at 6 months vs. Baseline Avg	9.36 (0.034)
CV at 6 months vs. Baseline CV	1.73 (0.027)
Kmax at 6 months vs. Baseline Kmax	−0.35 (0.042)
Kavg at 6 months vs. Baseline Kavg	−0.31 (0.048)

**Table 7 jcm-14-04489-t007:** Mean values and standard deviation (SD) in stage III KC before CXL, and after 1 and 6 months. NrCells = number of cells, CD = cell density, Avg = average cell surface, CV = cell variability, Hex = cell hexagonality, CCT = central corneal thickness, NrCellsPC = number of paracentral cells, CDPC = paracentral cell density, CVPC = paracentral cell variability, HexPC = paracentral hexagonality, CH = corneal hysteresis, CRF = corneal resistance factor, Kmin = minimum keratometry, Kmax = maximum keratometry, Kavg = average keratometry, and Astigm = astigmatism. Bold items signify statistically significant differences compared to the baseline value.

	Mean ± SD		Mean ± SD
Endothelial variables (both central and paracentral)			
NrCells	151.87 ± 36.4	NrCellsPC	1066.87 ± 267.2
NrCells I	156.12 ± 42.7	NrCellsPC I	1248.62 ± 420.3
NrCells6	137.5 ± 29.8	NrCellsPC6	1084.75 ± 238.5
CD	3017.62 ± 114.5	CDPC	2993.75 ± 53.6
CD I	3002.75 ± 1178	CDPC I	3009.75 ± 93.9
CD6	2971.5 ± 143.5	CDPC6	2984.87 ± 87.1
CV	26 ± 1.3	CVPC	27.75 ± 5.3
CV I	25.37 ± 3.7	CVPC I	26.62 ± 4.1
CV6	26.87 ± 3.6	CVPC6	27.37 ± 3.4
Hex	65.5 ± 3.6	HexPC	66.12 ± 2.6
Hex I	67 ± 3.5	HexPC I	67.75 ± 2.9
Hex6	65.5 ± 4.9	**HexPC6**	**68.5 ± 1.9**
CCT	494.37 ± 10	Avg	331.75 ± 13
CCT I	501.62 ± 14	Avg I	333.37 ± 13.4
CCT6	502.37 ± 23.7	Avg6	337.12 ± 16.1
Biomechanical corneal properties			
CH	8.62 ± 1.2	CRF	6.76 ± 1.6
CH I	8.04 ± 1.8	**CRF I**	**7.51 ± 1.3**
CH6	8.47 ± 2.4	CRF6	6.96 ± 2.9
Topographical properties			
Kmin	43.2 ± 1.9	Kmax	52.18 ± 2.7
Kmin I	43.35 ± 1.6	Kmax I	51.85 ± 3
Kmin6	43.45 ± 1.5	Kmax6	51.72 ± 3.5
Kavg	49.39 ± 3.2	Astigm	8.98 ±0.9
KavgI	47.60 ± 2.2	AstigmI	8.5 ± 1.8
Kavg6	49.39 ± 3.2	Astigm6	8.28 ± 2.3

**Table 8 jcm-14-04489-t008:** Significant differences in parameters before CXL, after CXL at 1 and 6 months, and *p* value of difference in stage III KC. HexPC = paracentral hexagonality; CRF = corneal resistance factor.

	Mean Difference (After CXL—Baseline) (*p* Value)
HexPC at 6 months vs. Baseline Hex	2.37 (0.004)
CRF at 1 month vs. Baseline CRF	0.75 (0.004)

**Table 9 jcm-14-04489-t009:** Mean values and standard deviation (SD) in stage IV KC before CXL, and after 1 and 6 months. NrCells = number of cells, CD = cell density, Avg = average cell surface, CV = cell variability, Hex = cell hexagonality, CCT = central corneal thickness, NrCellsPC = number of paracentral cells, CDPC = paracentral cell density, CVPC = paracentral cell variability, HexPC = paracentral hexagonality, CH = corneal hysteresis, CRF = corneal resistance factor, Kmin = minimum keratometry, Kmax = maximum keratometry, Kavg = average keratometry, and Astigm = astigmatism. Bold items signify statistically significant differences compared to the baseline value.

	Mean ± SD		Mean ± SD
Endothelial variables (both central and paracentral)			
NrCells	111 ± 24.08	NrCellsPC	954.25 ± 30.8
NrCells I	106 ± 36.6	NrCellsPC I	885.25 ± 59.3
NrCells6	78.5 ± 45.9	NrCellsPC6	878.25 ± 55.9
CD	3032 ± 156.6	CDPC	2809 ± 337.7
CD I	3100.250± 184.8	CDPC I	3002.25 ± 180.9
CD6	3018 ± 238.3	CDPC6	3044.25 ± 261.5
CV	28.75 ± 1.5	CVPC	34 ± 14.5
CV I	24.5 ± 4.6	CVPC I	40.5 ± 29.3
CV6	26.75 ± 9.8	CVPC6	38 ± 25.5
Hex	71.25 ± 7.4	HexPC	68.75 ± 4.5
Hex I	71 ± 5.2	HexPC I	69.5 ± 3.4
Hex6	72 ± 5.3	HexPC6	68.5 ± 4.7
Avg	330.5 ± 17.5	CCT	423.25 ± 36.4
Avg I	323.5 ± 20.5	CCT I	429.25 ± 40.5
Avg6	356.75 ± 14.8	CCT6	431 ± 46.8
Biomechanical corneal properties			
CH	7.9 ± 1.2	CRF	6.425 ± 1.1
CH I	7.77 ± 1.3	**CRF I**	**5.950** ± 0.9
CH6	7.25 ± 1.3	CRF6	5.875 ± 1.4
Topographical properties			
Kmin	54.59 ± 2.9	Kmax	63.045 ± 5.7
Kmin I	55.19 ± 4.5	Kmax I	63.078 ± 6.4
Kmin6	55.37 ± 2.6	Kmax6	62.483 ± 6.3
Kavg	58.82 ± 3.9	Astigm	9.568 ± 3.4
KavgI	59.14 ± 4.9	AstigmI	7.883 ± 5.1
Kavg6	58.92 ± 3.9	Astigm6	7.115 ± 5.6

**Table 10 jcm-14-04489-t010:** Significant differences in parameters before CXL, after CXL at 1 and 6 months, and *p* value of difference in stage IV KC. CRF = corneal resistance factor.

	Mean Difference (After CXL—Baseline) (*p* Value)
CRF at 1 month vs. Baseline CRF	0.475 (0.036)

**Table 11 jcm-14-04489-t011:** Significant correlations between topographical and endothelial corneal properties (before corneal collagen crosslinking).

Pair of Variables	Pearson Coefficient of Correlation r (*p* Value)
Paracentral cell variability—Kmin	0.329 (0.007)
Paracentral cell variability—Kmax	0.272 (0.027)
Paracentral cell variability—Kavg	0.294 (0.016)
Paracentral number of cells—Kmax	−0.305 (0.014)
Paracentral number of cells—Kmin	−0.323 (0.009)
Paracentral number of cells—Kavg	−0.329 (0.007)

**Table 12 jcm-14-04489-t012:** Significant correlations between topographical and biomechanical corneal properties (before corneal collagen crosslinking).

Pair of Variables	Pearson Coefficient of Correlation r (*p* Value)	Pair of Variables	Pearson Coefficient of Correlation r (*p* Value)
Corneal hysteresis—Kmax	−0.302 (0.014)	CCT—Kmax	−0.492 (<0.001)
Corneal hysteresis—Kmin	−0.329 (0.007)	CCT—Kmin	−0.535 (<0.001)
Corneal hysteresis—Kavg	−0.330 (0.007)	CCT—Kavg	−0.536 (<0.001)
Corneal resistance factor—Kmax	−0.504 (<0.001)	Corneal resistance factor—Kmin	−0.387 (0.001)
Corneal resistance factor—Kavg	−0.479 (<0.001)	Corneal resistance factor—Astigmatism	−0.385 (0.002)

**Table 13 jcm-14-04489-t013:** Significant correlations between age, topographical and biomechanical baseline corneal properties, and the degree of difference in endothelial parameters.

Pair of Variables	Pearson Correlation (*p* Value)
**Age—CV 1-0**	0.377 (0.002)
**Age—CV 6-0**	0.298 (0.015)
**CH—Avg 6-0**	−0.285 (0.02)
**CRF—CV 1-0**	−0.308 (0.012)
**Kavg—Avg 6-0**	0.349 (0.004)
**Kavg—CV 6-0**	0.250 (0.043)

**Table 14 jcm-14-04489-t014:** Significant correlations between baseline endothelial properties and the degree of difference in endothelial parameters.

Pair of Variables	Pearson Correlation (*p* Value)
**CV—NrCells 6-0**	−0.288 (0.019)
**CV—Avg 6-0**	0.248 (0.045)

## Data Availability

The datasets generated during and/or analyzed during the current study are available from the corresponding authors on reasonable request.

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
