# Peer review of "Impact of Corneal Crosslinking on Endothelial and Biomechanical Parameters in Keratoconus"

_jcm, 2025, doi:10.3390/jcm14134489_

Round 1
Reviewer 1 Report
Comments and Suggestions for Authors
- Were the patients on any medications between 2019 and 2022 to maintain endothelial cell density?
- What are the primary causes of endothelial cell density reduction at each stage of keratoconus, and how can endothelial cell density be preserved throughout the progression of the disease?
Reviewer 2 Report
Comments and Suggestions for Authors
Questions:
1. The study was designed as a non-randomised study. Please discuss its limitations in the discussion section so that someone can try a randomised study in the future.
2. In the biomechanical property, Intracocular pressure (IOP) was not discussed with the different Keratoconus of T0, T1 and T6?
